# FedSUV: Validity and Utility-guided Client Selection for Federated Learning

## Abstract

Federated Learning faces significant challenges arising from two critical uncertainties: the validity of a client's participation, which can be compromised by network and system heterogeneity, and the utility of the data contributed by each client, which varies due to heterogeneous statistical data. Traditional client selection methods often treat these uncertainties as a whole, leading to suboptimal performance. To address this issue, we propose FedSUV, an innovative client selection framework that decouples validity and utility uncertainties. FedSUV approaches client selection from a multi-objective optimization perspective, employing advanced bandit algorithms: a confidence bound-based linear contextual bandit for assessing validity and a Gaussian Process bandit for evaluating utility. We validate the effectiveness of FedSUV through both theoretical analysis and large-scale experiments conducted within our physical cluster.

## 1 Introduction

Federated learning (FL) is a distributed machine learning paradigm that trains models locally on multiple clients and only sends model updates to a central server for aggregation, thereby protecting data privacy Konečnỳ et al. (2015); McMahan et al. (2017). Fig. 2 shows the workflow of FL and highlights several challenges in its practical deployment. One challenge is communication limitations. The frequent exchange of model parameters between clients and the central server is constrained by limited bandwidth and channel capacity Li et al. (2020b); Shi et al. (2023). To address this issue, client selection—choosing a subset of clients to participate in the FL training—has been widely considered an effective method for optimizing FL.

Common barriers in client selection include network heterogeneity, system heterogeneity, and heterogeneous statistical data Ying et al. (2020); Tian et al. (2024). On one hand, local training on certain clients is likely to fail due to system heterogeneity, such as hardware constraints and runtime variance. Even if local training is successful, the model may fail to be transmitted during the aggregation process due to network heterogeneity, such as poor channel conditions and high latency. Consequently, a selected client may become an invalid participant, requiring additional communication rounds Abdelmoniem et al. (2023). We refer to the ability of a client to successfully participate within one round as *validity*. On the other hand, the contribution of each client to the overall FL system varies due to heterogeneous statistical data Yang et al. (2024), which we refer to as *utility*. Naturally, we aim to select clients with high utility in each round of FL. However, a client with high utility but low validity is not beneficial. The fundamental challenge is finding the right balance between validity and utility. Adding to the complexity, both these factors are inherently uncertain, highlighting the crucial need to adeptly address these uncertainties for optimal client selection.

This paper focuses on simultaneously managing these two types of uncertainty—validity and utility—in client selection. Existing works on client selection in FL often address the uncertainties by treating them as a single entity, as illustrated Fig. 1. For instance, Oort directly multiplies the two types of uncertainties and then employs a single Multi-Armed Bandit (MAB)

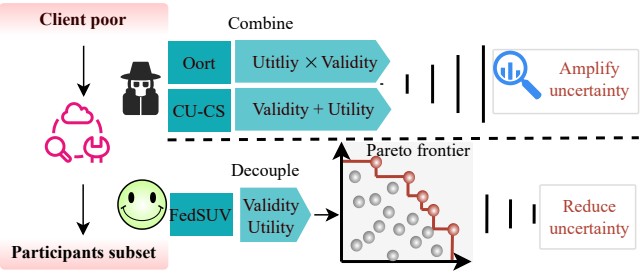

Figure 1: Motivation of FedSUV.

method for selection Lai et al. (2021). Similarly, CU-CS combines these uncertainties by addition and also uses a single MAB approach for client selection Shi et al. (2023). However, such methods

may amplify uncertainties rather than mitigate them Iqbal et al. (2023). Treating uncertainties as a combined metric, whether by addition or multiplication, overlooks the complex dynamics introduced by their interaction. This approach also fails to account for the unique contribution of each uncertainty factor to the overall metric. For instance, a high aggregated score might occur when low validity is compensated by high utility, thereby masking the deficiencies of individual uncertainty factors. Consequently, such coupling can reduce system robustness and lead to suboptimal performance, adversely affecting both convergence speed and accuracy in the FL system.

In this paper, we propose FedSUV, a new and efficient framework for client selection in FL. As illustrated in Fig. 1, FedSUV decouples the two uncertainties of validity and utility in FL, aiming to explicitly explore their interaction. Selecting participants from numerous clients can be modeled as a multi-armed bandit problem Lai et al. (2021). Consequently, FedSUV employs the confidence bound-based linear contextual bandit algorithm Li et al. (2010); Shi et al. (2023) to predict the expected validity of clients and applies the confidence bound-based Gaussian Process(GP) bandit algorithm Krause & Ong (2011); Liu et al. (2019); Li et al. (2010) to estimate the expected utility of clients. Leveraging these two estimates, FedSUV identifies the optimal subset of clients in each round. Specifically, it strives to select Pareto-optimal clients or clients near the Pareto-optimal front, drawing inspiration from the Pareto Learning Algorithm Zuluaga et al. (2013).

In summary, this paper makes the following contributions. First, we decouple the uncertainty factors in FL and address them from multi-objective MAB perspectives. Second, we introduce a novel method that effectively balances validity and utility by employing a dual-model approach—training both a linear model and a GP model to tackle the distinct uncertainties in FL. Furthermore, we establish an analytical framework that demonstrates FedSUV's performance, showing it effectively captures and reduces uncertainty and achieves a sublinear regret bound with high probability, ensuring rapid convergence from a theoretical standpoint. Meanwhile, FedSUV speeds up convergence by up to $3.54\times$ and improves final accuracy by up to 30.54% in extensive cluster experiments.

## 2 RELATED WORK

Fig. 2 illustrates a typical FL framework and highlights the challenges encountered in real-world implementations. Effective client selection is crucial to addressing these challenges. Client selection methods can be broadly categorized into three approaches: heuristic-based selection, reinforcement learning-based selection, and bandit-based selection.

Traditional client selection methods predominantly rely on heuristics, evaluating factors such as data resources (e.g., FedCS Nishio & Yonetani (2019), MCFL Li et al. (2020a), Harmony Tian et al. (2022)) and distinct training processes (e.g., SAFA Wu et al. (2020), REFL Abdelmoniem et al. (2023)). However, heuristic-based approaches may lack robustness in new or unseen scenarios, often requiring extensive domain expertise and tuning.

Reinforcement learning-based solutions, such as FAVOR Wang et al. (2020), FedMarl Zhang et al. (2022), and FedL Su et al. (2022), utilize reinforcement learning techniques for client selection. However, these methods can incur additional training time overhead due to the complexity of deep reinforcement learning and lack theoretical guarantees.

Bandit-based solutions present a promising alternative. For instance, Xia et al.2020 demonstrated that MAB-based methods can achieve significantly lower training loss compared to various benchmarks. Lai et al.2021 introduced a framework that prioritizes client participation based on the utility of existing data to enhance model accuracy and training speed, treating client selection as a standard MAB problem without incorporating contextual features. Huang et al.2022a proposed a contextual MAB-based client selection algorithm for mobile edge computing systems, considering computation and communication times as context. Shi et al.2023 designed a client selection scheme based on MAB for volatile FL, optimizing convergence rates. MAB-based methods offer theoretical guarantees and can be adapted for exploration, making them a valuable approach for reference. Collectively, these approaches address FL uncertainties from a holistic perspective, yet they often overlook the benefits of decoupling. In this paper, we leverage multi-objective MAB principles to enhance client selection in FL. Our approach aims to decouple and balance two critical factors—validity and utility—through a refined selection process, thereby improving the overall performance and efficiency of client selection.

Figure 2: Workflow and challenges of Federated Learning.

## 3 PROBLEM FORMULATION

In this section, we present the fundamental problem formulation for client selection in FL. By observing the workflow in FL, as depicted in Fig. 2, we identify two types of uncertainty—validity and utility—to formulate the optimization problem.

Formally, consider a client pool with $N$ clients denoted by $\mathcal{N} = \{1, 2, \cdots, N\}$. To indicate whether a candidate client $i \in \mathcal{N}$ is selected as a participant by the FL server at the beginning of each round $t \in \mathcal{T} = \{1, 2, \cdots, T\}$, we define the following binary variable:

$$\xi_{i,t} = \begin{cases} 1, & \text{if the FL server selects client } i \text{ at round t,} \\ 0, & \text{otherwise.} \end{cases} \quad (1)$$

Due to communication limitations, the FL sever can select a maximum of $K$ participants from the client pool $\mathcal{N}$, i.e., $\sum_{i=1}^{N} \xi_{i,t} \leq K$. We define $v_{i,t}$ to indicate whether client $i$ is a valid participant:

$$v_{i,t} = \begin{cases} 1, & \tau_{i,t} \leq T_d, \\ 0, & \tau_{i,t} > T_d, \end{cases} \quad (2)$$

where $\tau_{i,t}$ represents the total time spent by client $i$ in round $t$, and $T_d$ is the maximum allowable duration for each round to prevent indefinite delays caused by potential stragglers. Suppose that for each $i$ and $t$, $v_{i,t}$ follows an unknown fixed underlying distribution, with uncertainty primarily arises from fluctuations in running times across rounds Lai et al. (2021). More precisely, there is some fixed but unknown $v_i$ such that $v_i = \mathbb{E}[v_{i,t}]$.

Similarly, let $u_{i,t}$ denote the utility of client $i$ at round $t$ for the overall FL system. There is some fixed but unknown $u_i$ such that $u_i = \mathbb{E}[u_{i,t}]$, where uncertainty in utility mainly arises from the computation of stochastic gradient descent on different minibatches during client training McCandlish et al. (2018); Chen et al. (2023). Given that client utility is heavily influenced by participant validity, our global optimization goal is to maximize the cumulative client utility while considering validity over $T$ rounds, which can be formulated as:

$$\max \sum_{t=1}^{T} \sum_{i=1}^{N} \xi_{i,t} \cdot v_{i,t} \cdot u_{i,t}, \quad \text{s.t.} \sum_{i=1}^{N} \xi_{i,t} \leq K, \ t \in \mathcal{T}. \quad \text{(OPT)}$$

The problem we aim to tackle involves selecting the optimal set of clients amidst two forms of uncertainties, highlighting the need for efficient capture and mitigation of these uncertainties.

## 4 FEDSUV: FRAMEWORK DESIGN

In this section, we describe the design details of FedSUV. We start by explaining how FedSUV quantifies client utility (§4.1), then transform the client selection objective using multi-objective optimization principles (§4.2). Next, we describe how FedSUV estimates the uncertainty associated with client's validity and utility (§4.3). Finally, we illustrate the online learning process (§4.4).

### 4.1 DESIGNING CLIENT UTILITY

Accurately quantifying client utility is crucial, as it reflects the extent to which the client's training contributes to improving the global model's accuracy. Therefore, an ideal design of client utility should effectively capture the client's statistical data to enhance model performance for various training tasks. Drawing inspiration from the principles of importance sampling Katharopoulos & Fleuret (2018); Zhao & Zhang (2015), we consider two key factors in this quantification:

- **Accumulate training loss**: $L_{i,t}$. This factor is defined as $L_{i,t} = |B_i|\sqrt{\frac{1}{|B_i|}\sum_{k\in B_i} Loss(k)^2}$, where $B_i$ represents the batch of data from client $i$ and $Loss(k)$ is the loss for sample $k$. Clients with higher accumulated loss are considered more important for future rounds because their data can provide significant learning opportunities for the model Lai et al. (2021), thus enhancing data exposure and improving generalization.

- **Incremental improvement in training accuracy**: $\Delta_{i,t}$. In addition to the commonly used accumulated training loss factor, we introduce this factor to provide a more comprehensive quantification of a client's incremental contribution. This factor measures the model's accuracy increase in the current training round. Incremental accuracy improvements indicate the marginal benefits of a client in the joint training. Therefore, a small incremental improvement suggests either the model has already effectively learned from the client's data or the quality of the local data is unsatisfactory, and thus may not provide substantial additional value.

These two factors ensure prioritization of clients that contribute most to improving global model performance. Specifically, the incremental improvement in training accuracy ensures clients offering immediate benefits are selected, while the accumulated training loss highlights those whose data remains underutilized and valuable for future training. Furthermore, we design the utility as follows:

$$u_{i,t} = L_{i,t} \cdot \Delta_{i,t}. \tag{3}$$

By multiplying these two factors, the utility metric accounts for both immediate impact on accuracy and potential for future learning. This product-based utility metric prevents the selection of clients that have a high loss but negligible incremental improvement, or vice versa, thus optimizing the selection process to maximize the overall model performance.

## 4.2 TRANSFORMING OBJECTIVE

When optimizing client selection, we aim to select clients with high expected validity and utility, necessitating a delicate balance between these two goals. Inspired by Zuluaga et al. (2016); Garivier et al. (2024), we address this balancing issue using the principles of multi-objective optimization.

Formally, each client $i$ is associated with a $d$-dimensional feature vector $\boldsymbol{x}_i$, which includes attributes such as CPU FLOPS, CPU cores, Memory, GPU FLOPS, GPU count, and Data size. The set of all client feature vectors is denoted as $\mathcal{X} = \{\boldsymbol{x}_i | i \in \mathcal{N}\}$. The expected validity and utility of each client are modeled as the functions of their feature vector, specifically $v_i = v(\boldsymbol{x}_i)$ and $u_i = u(\boldsymbol{x}_i)$.

For convenience, we define the combined objective function as: $h(\boldsymbol{x}) = \begin{pmatrix} v(\boldsymbol{x}) \\ u(\boldsymbol{x}) \end{pmatrix}$, where $\boldsymbol{x} \in \mathcal{X}$.

While there is no best evaluation standard from a multi-objective optimization perspective, we are interested in identifying Pareto-optimal solutions Zuluaga et al. (2013). Formally, we define Pareto-optimality using the canonical partial order:

**Definition 1.** *For any two clients $i$ and $j$, $h(\boldsymbol{x}_j) \preceq h(\boldsymbol{x}_i)$ if and only if $v(\boldsymbol{x}_j) \leq v(\boldsymbol{x}_i)$ and $u(\boldsymbol{x}_j) \leq u(\boldsymbol{x}_i)$.*

**Definition 2.** *A client $i$ is dominated by client $j$ if and only if $h(\boldsymbol{x}_i) \preceq h(\boldsymbol{x}_j)$.*

**Definition 3.** *A client $i$ is called Pareto-optimal if no client $j$ exists that dominates it.*

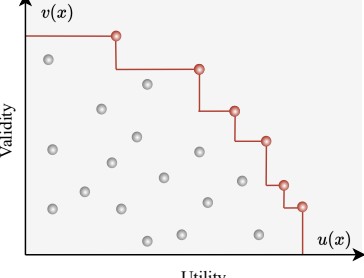

Figure 3: Pareto-optimality example.

Fig. 3 shows an example of Pareto-optimality, where each point represents a client based on expected validity and utility. The optimal clients are on the frontier and our objective is to identify and select clients that are either Pareto-optimal or close to the Pareto-optimal frontier.

## 4.3 ESTIMATING THE UNCERTAINTY

The primary obstacle in the client selection problem (OPT) lies in the integration of the highly variable uncertainty into the unknown expectations of client validity and utility. Therefore, estimating the two types of uncertainty is crucial. In fact, the process of client selection in FL can be viewed as a MAB problem Huang et al. (2020; 2022b); Xia et al. (2021); Wei et al. (2021); Xia et al. (2020);

Lai et al. (2021), which is a type of sequential decision problem Auer (2002). Thus, we propose using bandit methods to estimate these uncertainties.

**Estimating the validity.** Linear model are effective for capturing the relationship between client features and performance, making them suitable for client selection tasks Shi et al. (2023). Inspired by LinUCB Li et al. (2010); Abbasi-Yadkori et al. (2011); Shi et al. (2023), we assume the expected validity $v_i$ of a client $i$ is a linear function with an unknown parameter vector $\boldsymbol{\theta}$, i.e., $v_i = v(\boldsymbol{x}_i) = \boldsymbol{\theta} \cdot \boldsymbol{x}_i + \eta_i$, where $\eta_i$ represents random noise.

Based the ridge regression, we can obtain an interval estimate of $v_i$ with probability at least $1 - \delta$, given by

$$v(\boldsymbol{x}_i) \in \left[ \boldsymbol{\theta}_t \cdot \boldsymbol{x}_i - \alpha \|\boldsymbol{x}_i\|_{\boldsymbol{H}_t^{-1}}, \ \boldsymbol{\theta}_t \cdot \boldsymbol{x}_i + \alpha \|\boldsymbol{x}_i\|_{\boldsymbol{H}_t^{-1}} \right], \tag{4}$$

where $\delta \in (0, 1)$, $\alpha = 1 + \sqrt{\log(4/\delta)/2}$, $\boldsymbol{H}_t$ is the feature covariance matrix, and $\boldsymbol{\theta}_t$ is the estimate of $\boldsymbol{\theta}$ obtained from LinUCB at round $t$.

**Estimating the utility.** FedSUV employs a GP model to estimate utility, leveraging its strong ability to address exploration-exploitation tradeoffs in online learning of complex systems Liu et al. (2019; 2022), as well as its robustness in handling noise Krause & Ong (2011); Rasmussen (2003). Given this GP model, which is characterized by a mean function $\mu_{t-1}(\boldsymbol{x}_i)$ and a covariance function $\sigma_{t-1}(\boldsymbol{x}_i)$, we can obtain an interval estimate of $u_i$ with probability at least $1 - \delta$, provided by

$$u(\boldsymbol{x}_i) \in \left[ \mu_{t-1}(\boldsymbol{x}_i) - \beta_t^{1/2} \sigma_{t-1}(\boldsymbol{x}_i), \mu_{t-1}(\boldsymbol{x}_i) + \beta_t^{1/2} \sigma_{t-1}(\boldsymbol{x}_i) \right], \tag{5}$$

where $\delta \in (0, 1)$ and $\beta_t = 2 \log \left( \frac{|\mathcal{X}| \pi^2 t^2}{3\delta} \right)$.

## 4.4 ONLINE LEARNING PROCESS

In this part, we detail how FedSUV selects clients in each round. FedSUV utilizes a dual-model approach, training both a linear model and a GP model to address uncertainties in FL. The combined approach is represented using the following rectangle, derived from Eq. (4) and Eq. (5):

$$Q_t(\boldsymbol{x}) = \left\{ \begin{pmatrix} v \\ u \end{pmatrix} \middle| \begin{pmatrix} \boldsymbol{\theta}_t \cdot \boldsymbol{x} - \alpha \|\boldsymbol{x}\|_{\boldsymbol{H}_t^{-1}} \\ \mu_{t-1}(\boldsymbol{x}) - \beta_t^{1/2} \sigma_{t-1}(\boldsymbol{x}) \end{pmatrix} \preceq \begin{pmatrix} v \\ u \end{pmatrix} \preceq \begin{pmatrix} \boldsymbol{\theta}_t \cdot \boldsymbol{x} + \alpha \|\boldsymbol{x}\|_{\boldsymbol{H}_t^{-1}} \\ \mu_{t-1}(\boldsymbol{x}) + \beta_t^{1/2} \sigma_{t-1}(\boldsymbol{x}) \end{pmatrix} \right\}. \tag{6}$$

This rectangle provides an interval estimate for both validity and utility, guiding the client selection process.

The pseudocode of FedSUV is outlined in Appendix A. Each round of the algorithm consists of three main stages: Eliminating, Classifying, and Selecting. These stages are described as follows.

**Eliminating.** Including low-validity clients in the joint training can introduce negative bias to the global model and hurt final model accuracy. Despite their potential utility, such clients do not contribute effectively to the overall system. Therefore, to ensure effective client selection, it is necessary to avoid clients with low validity. Inspired by Shi & Shen 2021, FedSUV identifies an elimination set $\mathcal{E}_t$ each round using the confidence bound:

$$\mathcal{E}_t = \left\{ \boldsymbol{x}_i \in \mathcal{X}_{t-1} \ \middle| \ \boldsymbol{\theta}_t \cdot \boldsymbol{x}_i + \alpha \|\boldsymbol{x}_i\|_{\boldsymbol{H}_t^{-1}} \leq \max_{\boldsymbol{x}_i \in \mathcal{X}_{t-1}} \boldsymbol{\theta}_t \cdot \boldsymbol{x}_i - \alpha \|\boldsymbol{x}_i\|_{\boldsymbol{H}_t^{-1}} \right\}. \tag{7}$$

This set $\mathcal{E}_t$ consists of clients likely to have low validity. FedSUV then eliminates these clients from the candidate pool, updating it to $\mathcal{X}_t = \mathcal{X}_{t-1} \setminus \mathcal{E}_t$, where $\mathcal{X}_t$ represents the set of candidate clients in round $t$. To control the proportion of eliminated clients, FedSUV sets a maximum deletion ratio $\rho$. If $|\mathcal{X}_{t-1}| \geq (1 - \rho)|\mathcal{X}_1|$, the elimination process proceeds. In our experiments, this ratio $\rho$ is typically set to 0.4.

**Classifying.** FedSUV infers whether a client is Pareto-optimal based on Eq. (6). For ensuring that all uncertainty regions are non-increasing with $t$, FedSUV refine this uncertainty region:

$$R_t(\boldsymbol{x}) = R_{t-1}(\boldsymbol{x}) \cap Q_t(\boldsymbol{x}), \tag{8}$$

where $R_0(\boldsymbol{x}) = \mathbb{R}^2$. Within $R_t(\boldsymbol{x})$, each client has an optimistic outcome $\max R_t(\boldsymbol{x})$ and a pessimistic outcome $\min R_t(\boldsymbol{x})$, both of which are determined in the partial order $\preceq$ and are unique. We denote the elements of $\min R_t(\boldsymbol{x})$ and $\max R_t(\boldsymbol{x})$ as follows:

$$\begin{pmatrix} \hat{v}_t(\boldsymbol{x}) \\ \hat{u}_t(\boldsymbol{x}) \end{pmatrix} := \min R_t(\boldsymbol{x}), \begin{pmatrix} \check{v}_t(\boldsymbol{x}) \\ \check{u}_t(\boldsymbol{x}) \end{pmatrix} := \max R_t(\boldsymbol{x}). \tag{9}$$

If there exists $\boldsymbol{x}' \neq \boldsymbol{x}$ such that $\max R_t(\boldsymbol{x}) \preceq \min R_t(\boldsymbol{x}')$, then the client corresponding to $\boldsymbol{x}$ is classified as a non-Pareto optimal client. FedSUV also removes these clients from the candidate client set $\mathcal{X}_t$, ensuring they will not be selected in the following rounds. Fig. 4 illustrates an example of eliminating and classifying at round $t$.

**Selecting.** After classification, FedSUV selects clients based on their uncertainty regions $R_t(\boldsymbol{x})$. The first step is to compute the length of the diagonal of the uncertainty region for each client:

$$w_t(\boldsymbol{x}) = \max_{\forall y, y' \in R_t(\boldsymbol{x})} \|y - y'\|. \tag{10}$$

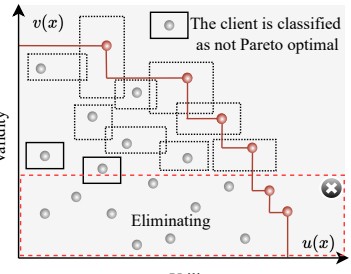

FedSUV then selects the client with the largest $w_t(\boldsymbol{x})$, which is determined by:

$$\boldsymbol{x}_t = \arg\max_{\boldsymbol{x} \in \mathcal{X}_t} w_t(\boldsymbol{x}). \tag{11}$$

Figure 4: Online learning process.

We refer $w_t(\boldsymbol{x}_t)$ as $w_t$ for simplicity. Intuitively, this rule biases the sampling toward exploring points most likely to be Pareto-optimal, thereby improving the overall performance.

For the remaining $K - 1$ selections, FedSUV draws inspiration from the UCB algorithm Srinivas et al. (2010). It selects the top $K - 1$ clients with the highest utility within their uncertainty regions, specifically those with the highest $\check{u}_t(\boldsymbol{x})$ from the set $\mathcal{X}_t \setminus \{\boldsymbol{x}_t\}$. This process yields the selection strategy $\mathcal{A}_t$ for round $t$.

## 5 THEORETICAL RESULTS

In this section, we provide a detailed analysis of FedSUV's theoretical performance. We start by examining the elimination process, which ensures that clients with lower validity are systematically excluded, thereby improving the resource efficiency of the client selection. Additionally, we show that clients classified as non-Pareto-optimal can be excluded from future rounds. Next, we discuss how FedSUV reduces uncertainty in client selection over time, leading to more accurate and reliable choices. Finally, we assess the regret performance of FedSUV, demonstrating that the algorithm quickly converges to an optimal solution. Collectively, these theoretical insights underscore the robustness and efficiency of FedSUV. Detailed proofs are provided in the Appendix B .

### 5.1 ELIMINATION ANALYSIS

Using Eq. (7) to eliminate clients means that the upper confidence bound of a deleted client's validity is lower than the lower confidence bound of the best client's validity. Let $\boldsymbol{x}_{m_t}$ be defined as:

$$\boldsymbol{x}_{m_t} = \max_{\boldsymbol{x}_i \in \mathcal{X}_{t-1}} \boldsymbol{\theta}_t \cdot \boldsymbol{x}_i - \alpha \|\boldsymbol{x}_i\|_{\boldsymbol{H}_t^{-1}}. \tag{12}$$

And the expected validity of corresponding client is denoted as $v_{m_t}$. Theorem 5.1 demonstrates that a client with low validity will be eliminated with high probability.

**Theorem 5.1.** *For any $\delta \in (0, 1)$, if a client $i$ with a low expected validity value satisfies*

$$v_i \leq v_{m_t} - 2\alpha \left( \|\boldsymbol{x}_{m_t}\|_{\boldsymbol{H}_t^{-1}} + \|\boldsymbol{x}_i\|_{\boldsymbol{H}_t^{-1}} \right), \tag{13}$$

*then client $i$ will be eliminated with probability at least $1 - \frac{\delta}{2}$.*

Furthermore, Theorem 5.2 shows that non-Pareto-optimal clients can be excluded from the selection in subsequent rounds.

**Theorem 5.2.** *If a client is classified as non-Pareto-optimal, it is unnecessary to include it in the optimal selection process for the subsequent round.*

### 5.2 UNCERTAINTY REDUCTION

In this subsection, we first demonstrate that with probability at least $1 - \delta$, $\begin{pmatrix} v(\boldsymbol{x}) \\ u(\boldsymbol{x}) \end{pmatrix}$ falls within the uncertainty region $R_t(\boldsymbol{x})$ for all $\boldsymbol{x} \in \mathcal{X}$, as stated in Theorem 5.3 .

**Theorem 5.3.** *Given $\delta \in (0, 1)$, the following holds with probability at least $1 - \delta$:*

$$\begin{pmatrix} v(\boldsymbol{x}) \\ u(\boldsymbol{x}) \end{pmatrix} \in R_t(\boldsymbol{x}), \; \forall \boldsymbol{x} \in \mathcal{X}, t \geq 1. \tag{14}$$

Theorem 5.3 guarantees that the uncertainty region we build can capture the true expected validity and utility of clients. Moreover, since the selection rules used by FedSUV guarantee that the range of uncertainty decreases over time, meaning that $w_t$ decreases with $t$, we derive a bound for $w_t$, as detailed in Theorem 5.4 .

**Theorem 5.4.** *Given $\delta \in (0, 1)$, $\alpha = 1 + \sqrt{\log(4/\delta)/2}$, and $\beta_t = 2 \log \left( \frac{|\mathcal{X}|\pi^2 t^2}{3\delta} \right)$, the following inequality holds:*

$$\mathbb{P} \left\{ w_T \leq \sqrt{d\alpha^2 C_T/T + \beta_T \gamma_T C/T} \right\} \geq 1 - \delta, \tag{15}$$

*where $C_T = 2 \log \left( 1 + T/d \right)$, $C = 2/\log(1 + \sigma^{-2})$, and $\gamma_T = \mathcal{O}\left( (\log T)^{d+1} \right)$.*

### 5.3 REGRET PERFORMANCE

A fundamental difference between single- and multi-objective MAB is that, for the latter, it is not immediately clear which metric to use for evaluating regret. Similar to Zuluaga et al. 2013, we employ the following metric: $\text{Regret}_T = \sum_{t=1}^{T} V(\mathcal{A}^*) - V(\mathcal{A}_t)$, where $V(A)$ denotes the area in the objective space covered by the set $A$, defined as: $\bigcup_{a \in A} \{ y \in \mathbb{R}^2 \mid 0 \preceq y \preceq a \}$. Here, $\mathcal{A}^*$ denotes the set of optimal actions. The regret is measured by the difference in the area between the Pareto-optimal solution and the solution achieved by FedSUV over time.

Therefore, Theorem 5.5 demonstrates that FedSUV rapidly converges to an optimal solution.

**Theorem 5.5.** *Given $\delta \in (0, 1)$, $\epsilon = \min_{\substack{\forall v(\boldsymbol{x}) < v(\boldsymbol{x}') \\ u(\boldsymbol{x}) < u(\boldsymbol{x}')}} \min\{v(\boldsymbol{x}') - v(\boldsymbol{x}), u(\boldsymbol{x}') - u(\boldsymbol{x})\}$, $\alpha = 1 + \sqrt{\log(4/\delta)/2}$, and $\beta_t = 2 \log \left( \frac{|\mathcal{X}|\pi^2 t^2}{3\delta} \right)$, the following inequality holds with probability $1 - \delta$:*

$$Regret_T \leq \frac{4K(\alpha^2 dC_T + \beta_T \gamma_T C)}{\epsilon^2}, \tag{16}$$

*where $C_T = 2 \log \left( 1 + T/d \right)$, $C = \frac{2}{\log(1+\sigma^{-2})}$, and $\gamma_T = \mathcal{O}\left( (\log T)^{d+1} \right)$.*

## 6 EVALUATIONS

In this section, we comprehensively evaluate the performance of FedSUV across various popular datasets, comparing it against state-of-the-art methods to highlight its effectiveness in FL.To provide deeper insights into FedSUV's capabilities, we also conduct an in-depth analysis of its client selection process through simulation-based evaluations.

### 6.1 EXPERIMENTAL SETUP

**Infrastructure.** We utilize FedScale Lai et al. (2022) as the foundation for deploying and evaluating FL systems. FedSUV has been encapsulated into a module that integrates seamlessly with the global aggregation and client selection processes. Training and testing are conducted using a GPU cluster equipped with four NVIDIA A100 GPUs and Pytorch v1.13.1.

**Baselines.** FedSUV is compared with the following baselines to achieve a comprehensive performance evaluation:

- FedAvg McMahan et al. (2017): This method involves the central server randomly selecting a subset of available clients for participation in each training round, without considering other factors. Due to its simplicity, random selection is applicable to any FL scenario.
- SAFA Wu et al. (2020): This approach employs a compensatory selection strategy that favors clients with less frequent involvement to improve convergence rates under extreme conditions.
- REFL Abdelmoniem et al. (2023): REFL prioritizes clients with the least availability. Each client trains a local prediction model to estimate the probability of its future availability.
- Oort Lai et al. (2021): Oort utilizes a MAB-based strategy to assess client utility and prioritizes those clients that can offer the highest utility for global model training. This method takes into account both the importance of the training samples and the training time.

**Models and Datasets.** FedSUV and the baseline methods are evaluated across five types of training tasks: 1) training ResNet18 He et al. (2016) on CIFAR10 Krizhevsky (2009); 2) training MobileNet Sandler et al. (2018) on CIFAR10; 3) training ShuffleNet Zhang et al. (2018) on CIFAR100 Krizhevsky (2009); 4) training ResNet18 on FEMNIST Caldas et al. (2018); 5) training ResNet34 He et al. (2016) on GoogleSpeech Warden (2018). Detailed information about these tasks is provided in Table 1.

Table 1: Training tasks and their corresponding settings.

| Model | Dataset | LR | LE | BS | Clients | Samples |
|---|---|---|---|---|---|---|
| ResNet18 MobileNet | CIFAR10 | 0.01 | 1 | 10 | 3,500 | 60,000 |
| ShuffleNet | CIFAR100 | 0.04 | 1 | 20 | 3,500 | 60,000 |
| ResNet18 | FEMNIST | 0.05 | 5 | 20 | 3,500 | 805,263 |
| ResNet34 | GoogleSpeech | 0.04 | 1 | 16 | 2,618 | 105,829 |

LR: Learning Rate. LE: Local Epochs. BS: Batch Size.

**Parameter settings.** The regular configuration parameters for FL were set to their default values as provided by the FedScale framework, with no additional tuning applied. The parameter settings for the baseline methods align with those specified in the original studies McMahan et al. (2017); Wu et al. (2020); Lai et al. (2021); Abdelmoniem et al. (2023). Additionally, to ensure comparability, all other common hyperparameters across the FL methods were kept consistent. Unless otherwise specified, the round duration is set to 100 seconds, and 20 participants are selected in each round.

**Hardware Performance of Clients.** Consistent with Abdelmoniem et al. (2023), the feature vector for each client are extracted from AI Benchmark AIb (2021) and MobiPerf Mob (2021). Additionally, AI Benchmark provides training time data, while MobiPerf offers communication time details. This information enables us to generate the duration of each round for each client.

## 6.2 OVERALL PERFORMANCE

In this section, we assess the performance of FedSUV in model training and compare it with the aforementioned baselines. Besides the traditional Averaging methods employed for model aggregation McMahan et al. (2017), we also consider YoGi aggregation Reddi et al. (2021), which enhances model efficiency for the given participants. The main findings reveal that FedSUV significantly improves both the round-to-accuracy performance and the final model accuracy:
- With averaging aggregation, FedSUV accelerates model convergence by up to 3.54× and enhances final model accuracy by up to 30.54%.
- With YoGi aggregation, FedSUV speeds up model convergence by up to 2.92× and improves final model accuracy by up to 22.90%.

The following parts provide detailed insights into these improvement.

**Round-to-accuracy performance.** FedSUV significantly enhances round-to-accuracy performance compared to other methods, as shown in Table 2. For instance, with averaging aggregation, FedSUV achieves the target accuracy of 15% on the CIFAR100 dataset with ShuffleNet in just 130 rounds, which is 90 rounds faster than the next best method. Similarly, with YoGi aggregation, FedSUV reaches the target accuracy of 70% on FEMNIST with ResNet18 in

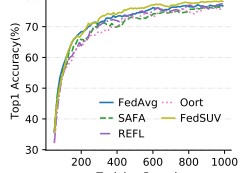 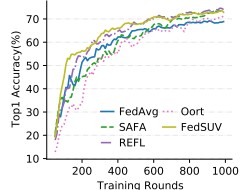

(a) Averaging aggregation   (b) YoGi aggregation

Figure 5: Test accuracy on FEMNIST with ResNet18 across various methods.

570 rounds, surpassing the second-best method by 20 rounds. Fig. 5 shows the convergence process on FEMNIST with ResNet18. Additional convergence figures are available in Appendix C.

Table 2: The required number of training rounds to achieve the same target accuracy.

| | Dataset—Model | CIFAR10—ResNet18 | | CIFAR10—MobileNet | | CIFAR100—ShuffleNet | | FEMNIST—ResNet18 | | GoogleSpeech—ResNet34 | |
|---|---|---|---|---|---|---|---|---|---|---|---|
| | Target accuracy | 45% | | 28% | | 15% | | 70% | | 50% | |
| | | # of Rounds | Speedup | # of Rounds | Speedup | # of Rounds | Speedup | # of Rounds | Speedup | # of Rounds | Speedup |
| | FedAvg | 500 | - | 880 | - | 460 | - | 200 | - | 570 | - |
| Averaging | SAFA | 580 | 0.86× | 500 | 1.76× | 220 | 2.09× | 210 | 0.95× | 500 | 1.14× |
| aggregation | Oort | 660 | 0.75× | 440 | 2× | 510 | 0.90× | 270 | 0.74× | 320 | 1.78× |
| | REFL | 550 | 0.90× | 410 | 2.15× | 320 | 1.44× | 240 | 0.83× | 310 | 1.84× |
| | FedSUV | **430** | **1.16×** | **390** | **2.26×** | **130** | **3.54×** | **200** | **1×** | **270** | **2.11×** |
| | FedAvg | 380 | - | 310 | - | 240 | - | 1000 | - | 600 | - |
| YoGi | SAFA | 290 | 1.31× | 430 | 0.72× | 320 | 0.75× | 600 | 1.67× | 530 | 1.13× |
| aggregation | Oort | 160 | 2.38× | 300 | 1.03× | 330 | 0.73× | 720 | 1.39× | 720 | 0.83× |
| | REFL | 290 | 1.31× | 270 | 1.15× | 220 | 1.09× | 590 | 1.69× | 570 | 1.05× |
| | FedSUV | **130** | **2.92×** | **170** | **1.82×** | **170** | **1.41×** | **570** | **1.75×** | **450** | **1.33×** |

These results highlight FedSUV's effectiveness in improving global aggregation by better client selection. FedSUV achieves this by decoupling the validity and utility uncertainties that affect selec-

tion. By capturing and reducing these uncertainties separately, FedSUV more efficiently identifies ideal clients, avoids short-sighted selections, and reaches target accuracy faster. In contrast, methods like Oort rely on a single, aggregated metric for evaluating clients, which overlooks the distinct types of variability and unpredictability in client contributions. This nuanced handling of uncertainties is what gives FedSUV its performance edge over other methods.

**Final model accuracy.** Table 3 shows that FedSUV consistently enhances final model accuracy across various datasets and models. For example, with averaging aggregation, FedSUV achieves the highest accuracy of 78.04% on the FEMNIST dataset with ResNet18, surpassing the worst-performing method, SAFA, by 2.60%. Similarly, with YoGi aggregation, FedSUV excels on the CIFAR10 dataset with ResNet18, reaching 60.82%, the highest accuracy among all

Table 3: The test accuracy (%) after convergence.

|  | Dataset Model | CIFAR10 ResNet18 | CIFAR10 MobileNet | CIFAR100 ShuffleNet | FEMNIST ResNet18 | GoogleSpeech ResNet34 |
|---|---|---|---|---|---|---|
| Averaging aggregation | FedAvg | 47.82 | 28.50 | 18.99 | 77.23 | 56.88 |
|  | SAFA | 48.69 | 30.90 | 24.75 | 76.06 | 60.19 |
|  | Oort | 46.69 | 31.01 | 19.32 | 76.70 | 60.12 |
|  | REFL | 48.05 | 31.12 | 22.00 | 77.66 | 58.88 |
|  | FedSUV | **49.22** | **33.94** | **24.79** | **78.04** | **60.28** |
| YoGi aggregation | FedAvg | 54.96 | 36.73 | 22.42 | 68.87 | 55.71 |
|  | SAFA | 55.19 | 35.25 | 19.13 | 72.87 | 53.64 |
|  | Oort | 59.51 | 35.75 | 22.49 | 71.27 | 54.14 |
|  | REFL | 54.80 | 36.06 | 22.08 | **73.60** | 54.90 |
|  | FedSUV | **60.82** | **39.43** | **23.51** | 73.01 | **56.00** |

methods. Notably, FedSUV achieves improvements of up to 28.31% across various datasets and models compared to Oort, which neglects the decoupling of uncertainties.

These improvements can be attributed to the design of the FedSUV framework. With more client selection rounds, FedSUV employs a dual-model approach-training both a linear model and a GP model to progressively refine the estimation of clients' validity and utility. This refinement leads to increasingly accurate estimations, enabling FedSUV to consistently select the most optimal clients and ensure sustained improvements in model accuracy.

### 6.3 SELECTION PROCESS ANALYSIS

To gain deeper insights into the client selection behavior of FedSUV, we conducted a simulated evaluation with 2500 clients. In this simulation, the validity and utility expectations of each client are uniformly distributed within the $[0, 1] \times [0, 1]$ interval. Our analysis focuses on the first 100 rounds of selection, with Fig. 6 illustrating the selection dynamics. Each block in the figure represents a client, with its position indicating the client's validity and utility expectations.

Fig. 6(a) illustrates the number of rounds each client was considered as a potential candidate clients $\mathcal{X}_t$. Notably, clients with validity expectations below 40% participated only in approximately 30 rounds, indicating that FedSUV effectively removed these clients through its elimination process. This outcome suggests that clients with lower validity expectations are progressively excluded, leading to a more refined selection of clients. The remaining clients, who

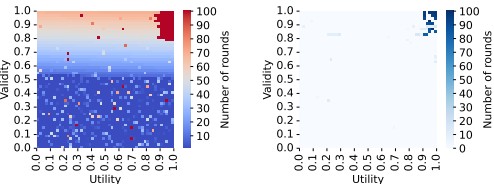

(a) Candidate participants rounds counted. (b) Selected participants rounds counted.

Figure 6: Statistics of client selection behavior.

were frequently considered as candidates, tend to be situated in the upper right corner of the plot, where both validity and utility expectations are high. This concentration indicates that FedSUV prioritizes clients that are Pareto-optimal or near Pareto-optimal.

Furthermore, Fig. 6(b) shows the cumulative number of times each client was selected. The pattern observed here mirrors that of the participation rounds: clients in the upper right corner, with high validity and utility expectations, were selected more frequently. This confirms that FedSUV effectively identifies and selects clients with superior performance metrics.

## 7 CONCLUSIONS

FedSUV represents a significant advancement in FL by introducing a dual-model approach from multi-objective MAB perspectives for client selection. By decoupling the uncertainties of validity and utility, FedSUV effectively mitigates the impact of uncertainty. Meanwhile, it utilizes both linear and GP models to intelligently capture and reduce these uncertainties, continuously identifying the optimal client group. Theoretical guarantees and empirical validation of FedSUV demonstrate its superiority and potential to revolutionize client selection strategies in FL.

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

## A    THE PSEUDOCODE OF FEDSUV

---
**Algorithm 1** FedSUV: Validity and Utility-guided Client Selection

---
1: **Input:** Maximum selection number $K$, deadline $T_d$, and deletion ratio $\rho$.
2: **Output:** Selection strategy $\{\mathcal{A}_t\}_{t \in \mathcal{T}}$.
3: Initialization;
4: **for** $t = 1$ to $T$ **do**
    —————————— **Eliminating** ——————————
5:    **if** $|\mathcal{X}_{t-1}| \geq (1 - \rho)|\mathcal{X}_1|$ **then**
6:        Determine the elimination set $\mathcal{E}$ according to Eq. (7);
7:        $\mathcal{X}_t = \mathcal{X}_{t-1} \setminus \mathcal{E}_t$;
8:    **end if**
    —————————— **Classifying** ——————————
9:    Compute $R_t(\boldsymbol{x}) = R_{t-1}(\boldsymbol{x}) \cap Q_t(\boldsymbol{x})$ for all $\boldsymbol{x} \in \mathcal{X}_t$;
10:    **for** all $\boldsymbol{x} \in \mathcal{X}_t$ **do**
11:        **if** $|\mathcal{X}_t| = K$ **then**
12:            **break;**
13:        **end if**
14:        **if** there exists $\boldsymbol{x}' \neq \boldsymbol{x}$ such that
15:        $\max R_t(\boldsymbol{x}) \preceq \min R_t(\boldsymbol{x}')$ **then**
16:            $\mathcal{X}_t = \mathcal{X}_t \setminus \{\boldsymbol{x}\}$;
17:        **end if**
18:    **end for**
    —————————— **Selecting** ——————————
19:    Choose a client based on Eq. (11);
20:    Choose the top $K - 1$ clients with the highest $\overset{\vee}{u}_t(\boldsymbol{x})$;
21:    Train the Linear model and GP model;
22: **end for**

---

## B    PROOFS OF THEORETICAL RESULTS

In this appendix, we provide proofs for the theorems that were omitted from the main paper.

*The proof of Theorem 5.1 .* Let $A$ denote the event $\left\{ \left| \boldsymbol{\theta}_t \cdot \boldsymbol{x}_i - v_i \right| \leq \alpha \|\boldsymbol{x}_i\|_{\boldsymbol{H}_t^{-1}}, \forall t \in \mathcal{T}, i \in \mathcal{N} \right\}$.
Then we have

$$\mathbb{P}(A) \geq 1 - \delta/2, \tag{17}$$

where $\delta \in (0, 1)$ and $\alpha = 1 + \sqrt{\log(4/\delta)/2}$ Li et al. (2010); Walsh et al. (2009). If event $A$ occurs, it implies

$$\left| \boldsymbol{\theta}_t \cdot \boldsymbol{x}_i - v_i \right| \leq \alpha \|\boldsymbol{x}_i\|_{\boldsymbol{H}_t^{-1}}, \text{ and } \left| \boldsymbol{\theta}_t \cdot \boldsymbol{x}_{m_t} - v_{m_t} \right| \leq \alpha \|\boldsymbol{x}_{m_t}\|_{\boldsymbol{H}_t^{-1}}. \tag{18}$$

Therefore, we can further derive that

$$\begin{aligned} \boldsymbol{\theta}_t \cdot \boldsymbol{x}_i + \alpha \|\boldsymbol{x}_i\|_{\boldsymbol{H}_t^{-1}} &\leq \boldsymbol{\theta} \cdot \boldsymbol{x}_i + 2\alpha \|\boldsymbol{x}_i\|_{\boldsymbol{H}_t^{-1}} \\ &\leq \boldsymbol{\theta} \cdot \boldsymbol{x}_i + 2\alpha \|\boldsymbol{x}_i\|_{\boldsymbol{H}_t^{-1}} + \boldsymbol{\theta}_t \cdot \boldsymbol{x}_{m_t} - v_{m_t} + \alpha \|\boldsymbol{x}_{m_t}\|_{\boldsymbol{H}_t^{-1}} \\ &= -\left( \boldsymbol{\theta} \cdot \boldsymbol{x}_{m_t} - \boldsymbol{\theta} \cdot \boldsymbol{x}_i - 2\alpha \|\boldsymbol{x}_i\|_{\boldsymbol{H}_t^{-1}} - 2\alpha \|\boldsymbol{x}_{m_t}\|_{\boldsymbol{H}_t^{-1}} \right) \\ &\quad + \boldsymbol{\theta}_t \cdot \boldsymbol{x}_{m_t} - \alpha \|\boldsymbol{x}_{m_t}\|_{\boldsymbol{H}_t^{-1}} \\ &\leq \boldsymbol{\theta}_t \cdot \boldsymbol{x}_{m_t} - \alpha \|\boldsymbol{x}_{m_t}\|_{\boldsymbol{H}_t^{-1}}, \end{aligned} \tag{19}$$

which implies that client $i$ is eliminated. $\qquad\square$

*The proof of Theorem 5.2 .* If a client with $\boldsymbol{x}$ is classified as non-Pareto-optimal, there exist at least one client with $\boldsymbol{x}''$ such that $\begin{pmatrix} v(\boldsymbol{x}) \\ u(\boldsymbol{x}) \end{pmatrix} \preceq \begin{pmatrix} v(\boldsymbol{x}'') \\ u(\boldsymbol{x}'') \end{pmatrix}$. Thus, the client with $\boldsymbol{x}$ can be ignored because if $\begin{pmatrix} v(\boldsymbol{x}') \\ u(\boldsymbol{x}') \end{pmatrix} \preceq \begin{pmatrix} v(\boldsymbol{x}) \\ u(\boldsymbol{x}) \end{pmatrix}$ then $\begin{pmatrix} v(\boldsymbol{x}') \\ u(\boldsymbol{x}') \end{pmatrix} \preceq \begin{pmatrix} v(\boldsymbol{x}'') \\ u(\boldsymbol{x}'') \end{pmatrix}$. This implies that $\boldsymbol{x}$ does not contribute

any advantage in the selection process compared to $\boldsymbol{x}''$, and therefore can be excluded in future rounds. □

*The proof of Theorem 5.3 .* According the proof of Theorem 5.1, we have

$$\mathbb{P}\Big\{\big|\boldsymbol{\theta}_t \cdot \boldsymbol{x}_i - v_i\big| \geq \alpha\|\boldsymbol{x}_i\|_{\boldsymbol{H}_t^{-1}}, \forall t \geq 1, i \in \mathcal{X}\Big\} \leq \frac{\delta}{2}. \tag{20}$$

In addition, we obtain:

$$\mathbb{P}\Big\{\big|f(\boldsymbol{x}_i) - \mu_{t-1}(\boldsymbol{x}_i)\big| \geq \beta_t^{1/2}\sigma_{t-1}(\boldsymbol{x}_i)\Big\} \leq e^{-\frac{\beta_t}{2}}. \tag{21}$$

This follows from the properties of the normal distribution: if $x \sim N(0,1)$ is drawn from a Gaussian distribution, then the upper bound of $Pr\{x \geq c\}$ is given by: $\mathbb{P}\{x \geq c\} \leq \frac{1}{2}e^{-c^2/2}$.

We conclude that the following holds with probability at most $|\mathcal{X}|e^{-\frac{\beta_t}{2}}$:

$$\big|f(\boldsymbol{x}_i) - \mu_{t-1}(\boldsymbol{x}_i)\big| \leq \beta_t^{1/2}\sigma_{t-1}(\boldsymbol{x}_i),$$
$$\forall \boldsymbol{x}_i \in \mathcal{X}. \tag{22}$$

By applying the union bound for $\boldsymbol{x}_i \in \mathcal{X}, t \geq 1$ and choosing $|\mathcal{X}|e^{-\frac{\beta_t}{2}} = \frac{3\delta}{\pi^2 t^2}$, we obtain the result of this theorem, noting that $\sum \frac{1}{t^2} = \frac{\pi^2}{6}$. □

*The proof of Theorem 5.4 .* Since $w_t$ is the maximum diagonal length of $R_t(\boldsymbol{x})$, we have

$$w_t^2 \leq \alpha^2\|\boldsymbol{x}_t\|_{\boldsymbol{H}_t^{-1}}^2 + \beta_t\sigma_{t-1}^2(\boldsymbol{x}_t), \tag{23}$$

which implies

$$\sum_{t=1}^{T} w_t^2 \leq \alpha^2\sum_{t=1}^{T}\|\boldsymbol{x}_t\|_{\boldsymbol{H}_t^{-1}}^2 + \sum_{t=1}^{T}\beta_t\sigma_{t-1}^2(\boldsymbol{x}_t), \tag{24}$$

Next, we will provide separate bounds for the two terms on the right-hand side of this inequality. We have

$$\sum_{t=1}^{T}\|\boldsymbol{x}_t\|_{\boldsymbol{H}_t^{-1}}^2 \leq 2\log\Big(\frac{\det(\boldsymbol{H}_T)}{\det(\boldsymbol{H})}\Big)$$
$$\leq 2\big(\log(\boldsymbol{H}_T) - \log\det(\boldsymbol{H})\big) \leq 2(d\log(1 + T/d) - d\log 1) \tag{25}$$
$$\leq 2d\log\big(1 + T/d\big).$$

The first inequality above is due to Lemma 11 in Abbasi-Yadkori et al. (2011), and the second-to-last inequality is due to the determinant-trace inequality in Abbasi-Yadkori et al. (2011).

Since $\beta_t$ is increasing, we have that

$$\sum_{t=1}^{T}\beta_t\sigma_{t-1}^2(\boldsymbol{x}_t) = \beta_T\sum_{t=1}^{T}\sigma^2(\sigma^{-2}\sigma_{t-1}^2(\boldsymbol{x}_t))$$
$$\leq \beta_T\sum_{t=1}^{T}\sigma^2\frac{\sigma^{-2}}{\log(1+\sigma^{-2})}\log\big(1 + \sigma^{-2}\sigma_{t-1}^2(\boldsymbol{x}_t)\big) \tag{26}$$
$$\leq \frac{1}{\log(1+\sigma^{-2})}\beta_T\sum_{t=1}^{T}\log\big(1 + \sigma^{-2}\sigma_{t-1}^2(\boldsymbol{x}_t)\big) \leq \frac{2}{\log(1+\sigma^{-2})}\beta_T\gamma_T.$$

The third-to-last inequality results from $\frac{s}{\log(1+s)} \leq \frac{\sigma^{-2}}{\log(1+\sigma^{-2})}$ for $s \in [0, \sigma^{-2}]$. The second-to-last inequality is due to Lemma 5.3 in Srinivas et al. (2010), and the maximum information gain can be bounded by $\gamma_T = \mathcal{O}\big((\log T)^{d+1}\big)$ Dani et al. (2008).

Combining Eqs. equation 25 and equation 26, we have

$$\sum_{t=1}^{T} w_t^2 \leq 2\alpha^2 d\log\big(1 + T/d\big) + \beta_T\gamma_T\frac{2}{\log(1+\sigma^{-2})}. \tag{27}$$

Using the Cauchy-Schwarz inequality, we obtain $\left(\sum_{t=1}^{T} w_t\right)^2 \leq T \sum_{t=1}^{T} w_t^2$, which leads to

$$\mathbb{P}\left\{\sum_{t=1}^{T} w_t \leq \sqrt{\alpha^2 dC_T T + \beta_T \gamma_T CT}\right\} \geq 1 - \delta. \tag{28}$$

Then, we have

$$\mathbb{P}\left\{\sum_{t=1}^{T} w_t/T \leq \sqrt{\alpha^2 dC_T/T + \beta_T \gamma_T C/T}\right\} \geq 1 - \delta. \tag{29}$$

Furthermore, due to the selection rule of FedSUV,
$w_t = \max_{\boldsymbol{x}\in\mathcal{X}_t} \max_{\forall y,y'\in R_t(\boldsymbol{x})} ||y - y'||$, we have $w_t = w_t(\boldsymbol{x_t}) \leq w_t(\boldsymbol{x_t}) \leq w_{t-1}(\boldsymbol{x_t}) \leq w_{t-1}$, implying that $w_t$ decreases with $t$. Therefore, $\sum_{t=1}^{T} w_t/T \geq w_T$, completing the proof. $\qquad\square$

*The proof of Theorem 5.5.* When $T \geq \frac{4(\alpha^2 dC_T + \beta_T \gamma_T C)}{\epsilon^2}$, we have $2\sqrt{\frac{\alpha^2 dC_T + \beta_T \gamma_T C}{T}} \leq \epsilon$. Due to Theorem 5.4, we know that $2w_T \leq \epsilon$. If $\begin{pmatrix} v(\boldsymbol{x}) \\ u(\boldsymbol{x}) \end{pmatrix} \preceq \begin{pmatrix} v(\boldsymbol{x}') \\ u(\boldsymbol{x}') \end{pmatrix}$, then

$$\begin{aligned}
\max R_T(\boldsymbol{x}) = \begin{pmatrix} \check{v}_T(\boldsymbol{x}) \\ \check{u}_T(\boldsymbol{x}) \end{pmatrix} &\preceq \begin{pmatrix} v(\boldsymbol{x}) + w_T \\ u(\boldsymbol{x}) + w_T \end{pmatrix} \preceq \begin{pmatrix} v(\boldsymbol{x}) + w_T + \hat{v}_T(\boldsymbol{x}') - v(\boldsymbol{x}') + w_T \\ u(\boldsymbol{x}) + w_T + \hat{u}_T(\boldsymbol{x}') - u(\boldsymbol{x}') + w_T \end{pmatrix} \\
&\preceq \begin{pmatrix} v(\boldsymbol{x}) - v(\boldsymbol{x}') + 2w_T + \hat{v}_T(\boldsymbol{x}') \\ u(\boldsymbol{x}) - u(\boldsymbol{x}') + 2w_T + \hat{u}_T(\boldsymbol{x}') \end{pmatrix} \preceq \begin{pmatrix} \hat{v}_T(\boldsymbol{x}') \\ \hat{u}_T(\boldsymbol{x}') \end{pmatrix} = \min R_T(\boldsymbol{x}'),
\end{aligned} \tag{30}$$

Thus, all clients that are not Pareto-optimal can be distinguished.

Consequently, after $\left\lceil \frac{4(\alpha^2 dC_T + \beta_T \gamma_T C)}{\epsilon^2} \right\rceil$ rounds, the remaining clients are Pareto optimal with probability $1 - \delta$. Additionally, assuming the valid value and utility both lie in the interval $[0, 1]$, and since $K$ clients are selected in each round, we have

$$\text{Regret}_T \leq \frac{4K(\alpha^2 dC_T + \beta_T \gamma_T C)}{\epsilon^2}, \tag{31}$$

which holds with probability $1 - \delta$. $\qquad\square$

## C    MORE CONVERGENCE FIGURES

Except the Fig. 5 in the main paper, we provide more convergence results as Figures 4, 5, 6 and 7. These results show that our method can accelerate FL training and obtain higher generalization performance.

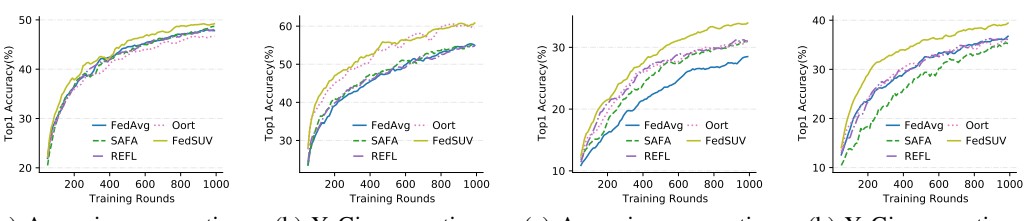

(a) Averaging aggregation    (b) YoGi aggregation    (a) Averaging aggregation    (b) YoGi aggregation

Figure 7:   Test accuracy on CIFAR10 with ResNet-18 across various methods.

Figure 8: Test accuracy on CIFAR10 with MobileNet across various methods.

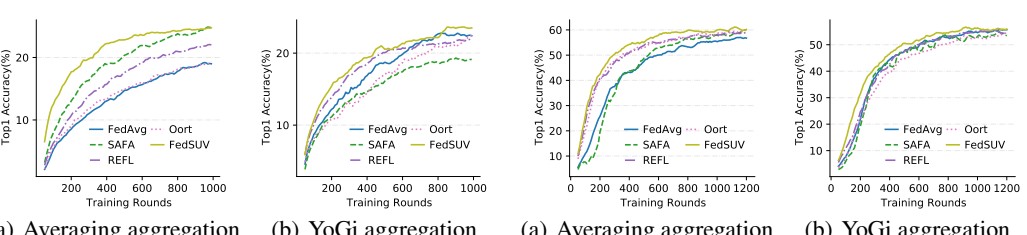

(a) Averaging aggregation    (b) YoGi aggregation    (a) Averaging aggregation    (b) YoGi aggregation

Figure 9: Test accuracy on CIFAR100 with ShuffleNet across various methods.

Figure 10:  Test accuracy on GoogleSpeech with ResNet34 across various methods.

## D    SOFTWARE AVAILABILITY

FedSUV has been integrated as a module within FedScale Lai et al. (2022), a framework designed for the emulation and evaluation of FL systems. The corresponding code can be found in the Supplementary Materials.

