# OpenReview forum: "FedSUV: Validity and Utility-guided Client Selection for Federated Learning"
_ICLR.cc/2025/Conference — ICLR 2025 Conference Withdrawn Submission_

### Official Review · Reviewer_sMa9 · 2024-10-24

**Soundness:** 2
**Presentation:** 3
**Contribution:** 2
**Rating:** 3
**Confidence:** 3

**Summary:**

This paper introduces the FedSUV algorithm to address the challenge of client selection in federated learning. The algorithm employs two key metrics—validity and utility—to capture network heterogeneity and data heterogeneity, respectively. Validity is assessed using a confidence-bound-based linear contextual bandit, while utility is evaluated through a Gaussian process bandit. The paper also includes numerical experiments to validate these claims.

**Strengths:**

1: This paper studies an interesting and important aspect, client selection, in federated learning.

2: This paper provides a theoretical regret analysis of their proposed method.

3: This paper compares with many other baseline algorithms.

**Weaknesses:**

I believe this work is valuable. However, the authors need to offer a more detailed justification of their model formulation. At present, the formulation appears to contradict the claims made in the introduction. Additionally, it would be highly beneficial if the authors considered incorporating fairness in client selection.

Major comments:

1: In the Introduction (line 55), the authors argue that "Treating uncertainties as a combined metric, whether by addition or multiplication, overlooks the complex dynamics introduced by their interaction". However, in their formulation and design (lines 145, 178), they also use multiplication without providing sufficient justification. What are the "complex dynamics introduced by their interaction"?

2: In line 182, the authors state, "This product-based utility metric prevents the selection of clients that have a high loss but negligible incremental improvement, or vice versa, thus optimizing the selection process to maximize the overall model performance." How does this metric handle clients with low loss but significant incremental improvement? Do such clients exist?

3: In line 257, the authors state "FedSUV then eliminates these clients from the candidate pool," referring to those with low validity. Will it also eliminate clients with high validity but low utility? This question stems from the Introduction (line 58), where the authors state,"For instance, a high aggregated score might occur when low validity is compensated by high utility, thereby masking the deficiencies of individual uncertainty factors."

4: In line 260, how and why is the ratio set to 0.4? Is this value optimal? Is it cherry-picked? Do authors select it from a subset of candidates?

5: Further justification is needed for Fig. 5, where FedSUV underperforms compared to REFL when using YoGi aggregation, but not averaging aggregation, in terms of final model accuracy.

6: In line 468, the authors state, "This outcome suggests that clients with lower validity expectations are progressively excluded, leading to a more refined selection of clients." How does the algorithm handle clients with low utility? Are there any effects due to the interaction between validity and utility?

7: In addition to the current related works, fairness in client selection is a natural question that arises. There is clearly existing literature on fairness-aware client selection in federated learning (see reference below). If fairness is not a focus of this study (which I believe it should be), the authors should provide a justification.

Shi, Yuxin, et al. "Fairness-aware client selection for federated learning." 2023 IEEE International Conference on Multimedia and Expo (ICME). IEEE, 2023.

Minor comment:

1: Typo in Fig. 1: "client pool"

**Questions:**

See weakness

---

### Official Review · Reviewer_nT3o · 2024-11-03

**Soundness:** 3
**Presentation:** 2
**Contribution:** 3
**Rating:** 5
**Confidence:** 3

**Summary:**

This paper studies the uncertainty in client selection in federated learning. Prior work in federated learning often combines the validity of the client's participation and the utility of the data contributed by each client as a whole, leading to suboptimal performances. Instead, the authors propose to separate these two factors and jointly learn them through multi-objective optimization. Using linear contextual bandits to evaluate validity and GP bandits for utility, the authors show that theoretically, FedSUV converges to an optimal solution with rate $O(K (\log T)^{d+1})$. The authors provided a set of empirical experiments to show that FedSUV outperforms previous methods in federated learning.

**Strengths:**

- The paper's approach to using multi-objective optimization for client selection based on validity and utility is novel.
- The proposed framework is clear and easy to follow.
- The empirical experiments support the paper's theoretical claims, showing a clear improvement over existing methods.

**Weaknesses:**

- Section 5 claims that FedSUV is both robust and effective. Which theorem shows the robustness of FedSUV?
- Figure 5 does not show an error bar and how many times the experiments were repeated.

**Questions:**

- What is the definition of $\sigma$ in Theorem 5.4?
- Is the regret in Theorem 5.5 optimal? Is there a lower bound on this regret?
- What is the size of $|\mathcal{X}|$? Can the authors quantify it in terms of $d$ or $N$?

---

### Official Review · Reviewer_3aUw · 2024-11-05

**Soundness:** 2
**Presentation:** 2
**Contribution:** 2
**Rating:** 3
**Confidence:** 3

**Summary:**

This submission focuses on client selection in federated learning. They focus on two metrics: validity and utility. The validity issue arises from the delay of communication and thus the server can't receive the local update in time. The utility arises from the heterogeneity of clients. The goal is to select clients that can contribute more, i.e., with higher utility.

They then model the client selection as an online learning problem. They propose a method FedSUV to eliminate, classify and select clients with theoretical guarantees.

**Strengths:**

The problem of client selection is very interesting.

**Weaknesses:**

I am not very familiar with the literature on client selection, but the modeling approach does not seem convincing to me.

- Equation (OPT) on page 3: I believe the goal of federated learning is to learn a model that performs well for every client, meaning it should minimize $\sum_{i} \text{Loss}(\text{model at client } i)$. It's unclear to me how this objective translates to the linear form of the objective in Eq. (OPT).
- Client utility in Eq. (3): I didn’t understand the definition of client utility in this context. Could you clarify how this utility connects to the goal of minimizing $\sum_{i} \text{Loss}(\text{model at client } i)$?
- Section 4.2: In this section, both validity and utility are modeled as functions of client features. What specific features are associated with clients? This setup doesn’t seem to align with real-world federated learning settings. I think in the real-world, each client is represented as a data set instead of a feature. Additionally, modeling validity as a linear function and utility via Gaussian processes seems unrealistic.

Overall, I feel the model diverges too much from real-world federated learning applications and lacks sufficient motivation.

**Questions:**

- In line 162, I don't understand why considering squared loss instead of other type of losses.
- In the experiments, for example, on cifar 10, what are your features $x_i$ for each client?

---

### Note · Authors · 2024-11-13

**Comment:**

Thank you for the reviewers' comments. After careful consideration, we believe that our paper may not be the fit for ICLR at this time, and we have therefore decided to withdraw our submission.

**Withdrawal Confirmation:**

I have read and agree with the venue's withdrawal policy on behalf of myself and my co-authors.